# L4Dog: Towards Robust BEV Perception for Quadruped Robots in Complex Urban Scenes

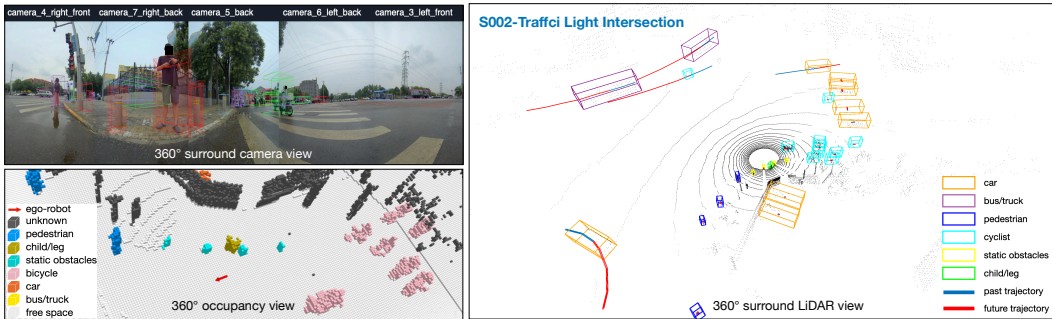

Figure 1: We introduce L4Dog, the first large-scale BEV perception dataset for quadruped robots in complex urban scenes. Featuring long-range perception in challenging scenarios, L4Dog provides high-quality manually annotated 3D ground truth and establishes benchmarks for multi-task BEV perception and occupancy prediction in 360° surrounding view.

## Abstract

Embodied intelligence in quadruped robots faces significant challenges in complex urban environments due to the limitations of traditional perception systems and the lack of comprehensive datasets for exteroceptive 3D perception. To address this, we introduce L4Dog, the first large-scale exteroceptive 3D perception dataset tailored for quadruped robots in open urban scenarios. L4Dog provides high-quality 360-degree surround-view sensor data and manual annotations, covering diverse urban scenes such as traffic-light intersections, open roads, subway station, etc. By formulating perception tasks as bird's-eye-view (BEV) space perception problems, we establish a multi-benchmark framework for BEV detection, tracking, trajectory prediction, and 3D traversable space occupancy estimation. The OmniBEV4D baseline method is proposed to unify multi-task perception (detection, tracking, prediction, and occupancy prediction) through shared temporal BEV features, enabling efficient and robust processing of dynamic urban environments. This work bridges the gap between current research and real-world deployment needs, offering a foundational resource for advancing autonomous navigation and decision-making in complex urban settings. The dataset will be made publicly available upon acceptance of this work.

## Introduction

Embodied intelligence, as a pivotal research direction in artificial intelligence, is accelerating the deployment of advanced AI technologies on robotic platforms. Among these, quadruped robots have emerged as ideal mobile platforms due to their exceptional terrain adaptability, high mobility, and flexibility, demonstrating broad application potential in scenarios such as visually impaired assistance, elderly mobility support, and last-mile delivery services. However, their practical deployment in open urban road environments faces significant challenges: unlike controlled indoor or campus settings, real-world urban roads feature complex environments with diverse road types and heterogeneous traffic participants (including pedestrians, vehicles, cyclists, static obstacles, etc.). Particularly under conditions of unpredictable traffic behaviors and highly dynamic environments, quadruped robots face heightened requirements for navigation planning, obstacle avoidance decisions, and interactive capabilities. Traditional forward-looking perception paradigms prove insufficient (Shah et al., 2021; 2022; Hirose et al., 2023), necessitating 360-degree surround-view and long-range

perception capabilities to identify fast-moving objects and enable evasive maneuvers; robustness in densely crowded pedestrian scenarios and occluded visibility conditions within complex urban areas must be enhanced; and critical advancements are required in establishing three-dimensional semantic traversability understanding to address challenges posed by road surfaces, curbs, tactile paving, and unknown obstacles.

As the core foundation for autonomous navigation and decision-making, perception systems in quadruped robots are typically divided into two categories: proprioceptive and exteroceptive perception (Miki et al., 2022). Proprioceptive perception focuses on processing sensor data from limb joints, foot contacts, and inertial measurement units (IMUs) – such as joint encoders and foot contact sensors – to estimate robot pose and control locomotion. Exteroceptive perception involves acquiring and interpreting external environmental information for object recognition and interaction. While extensive research in quadruped robotics has concentrated on terrain traversability optimization through proprioceptive enhancements (e.g., adaptability to varied terrains and parkour capabilities) (Miki et al., 2022; Hoeller et al., 2024; Cheng et al., 2024; Fink & Semini, 2020; Santana et al., 2024; Lin et al., 2023; Lee et al., 2020; Shi et al., 2023), exteroceptive perception in open-road scenarios remains critically underdeveloped: existing datasets are predominantly limited to small-scale or indoor/campus environments with low complexity and insufficient data quality to meet real-world urban road demands (Carlevaris et al., 2016; Yan et al., 2018; 2020; Hirose et al., 2018; Martin et al., 2021; Karnan et al., 2022; Hirose et al., 2023; Wang et al., 2024; Zhang et al., 2024; Luo et al., 2025). This pronounced gap between current research and future deployment needs highlights the urgency of establishing exteroceptive perception benchmarks tailored for complex urban environments.

Addressing this challenge, we present L4Dog, the first large-scale exteroceptive 3D perception dataset for quadruped robots in complex urban scenarios. The "L4" designation borrows from autonomous driving terminology, signifying level-4 autonomy in complex urban environments. Equipped with high-specification sensors enabling full 360-degree surround-view coverage, L4Dog surpasses existing quadruped datasets by encompassing challenging urban scenes including traffic-light intersections, open roads, subway stations, and tactile paving areas, featuring complex human-machine interaction scenarios with dense vehicle flows, pedestrians, and cyclists. We pioneer the formulation of outdoor quadruped perception tasks as surround-view bird's-eye-view (BEV) perception tasks, emphasizing three-dimensional BEV space perception for advanced autonomous navigation and decision-making. Our dataset provides high-quality 3D manual annotations, establishing multiple benchmark tasks including BEV detection, BEV tracking, and trajectory prediction. Furthermore, we introduce the first occupancy grid representation for 3D traversable space in quadruped robotics, with manual annotations of 360-degree occupancy grids surrounding the robot, thereby proposing the inaugural occupancy benchmark in exteroceptive perception for quadruped platforms. For multi-task perception (detection, tracking, prediction, occupancy prediction), we propose the OmniBEV4D baseline method, which formalizes exteroceptive tasks as BEV perception tasks and supports multi-task perception capabilities.

Our core contributions are summarized as follows:

1) We introduce L4Dog, the first BEV perception dataset for quadruped robots in open complex urban scenarios, featuring high-quality manual annotations. This work pioneers the formulation of quadruped exteroceptive 3D perception as a fused BEV-space perception task.

2) We establish a multi-benchmark framework for BEV environmental perception in quadruped robots, encompassing challenging tasks in BEV object detection, multi-target tracking, and trajectory prediction.

3) We propose the first occupancy network prediction framework for 360-degree 3D traversable space in quadruped robots, accompanied by high-quality occupancy annotations.

4) We develop the OmniBEV4D perception framework, which leverages shared temporal BEV features through a multi-task architecture to simultaneously enable BEV perception, tracking, trajectory prediction, and occupancy estimation, serving as the baseline method for L4Dog benchmark tasks.

The remainder of this paper is organized as follows: Section 2 reviews related datasets and perception methodologies in quadruped robotics; Section 3 details the L4Dog dataset; Section 4 presents the

multi-perception benchmarks and the OmniBEV4D baseline; Section 5 concludes with future work perspectives.

## 2 RELATED WORK

### 2.1 QUADRUPED ROBOT PERCEPTION DATASETS

Quadruped robot perception datasets can be categorized into proprioception (locomotion-focused) and exteroception (environmental understanding) types (Miki et al., 2022). This work focuses on exteroception, typically employing optical sensors such as RGB cameras, RGB-D cameras, and LiDAR. Notable datasets include SCAND (Karnan et al., 2022), which equipped ClearPath Jackal and Spot robots with 16-beam LiDAR and stereo RGB cameras to collect teleoperated traversal data for social navigation; NCLT (Carlevaris et al., 2016), which provides long-term campus data with 32-beam LiDAR and omnidirectional cameras for mapping applications; and RECON (Shah et al., 2021), ViKing (Shah et al., 2022), and GND (Liang et al., 2024), which serve as general-purpose mapping datasets. Specialized traversability datasets include ForestTrav (Ruetz et al., 2024), TRIP (Oh et al., 2024), and GoStanford (Hirose et al., 2018) for outdoor and indoor environments. Crucially, none of these datasets include explicit object recognition (e.g., pedestrian detection) or provide supervised annotations.

For explicit quadruped perception, FLOBOT (Yan et al., 2020) provides indoor pedestrian annotations using 16-beam LiDAR and stereo RGB-D cameras, while L-CAS (Yan et al., 2018) offers 3D pedestrian annotations in office environments with 16-beam LiDAR. QuadTrack (Luo et al., 2025) focuses on 2D multi-frame pedestrian tracking with panoramic cameras, and TBD Pedestrian (Wang et al., 2024) provides indoor pedestrian tracking with 3D annotations from single-beam LiDAR. Recent large-scale pedestrian datasets include JRDB (Martin et al., 2021), SiT (Bae et al., 2023), and CODa (Zhang et al., 2024), which feature varying sensor configurations with extensive 3D annotations.

Our work (L4Dog) belongs to explicit supervised quadruped perception, sharing similarities with (Bae et al., 2023), (Martin et al., 2021), and (Zhang et al., 2024) but introducing five key innovations. First, it represents the largest 3D-annotated quadruped exteroception dataset, being an order of magnitude larger than JRDB/CODa. Second, it captures Level 4 complex urban environments with high object density. Third, it pioneers the formulation of quadruped exteroception as 360°BEV perception. Fourth, it provides additional 3D occupancy annotations for traversable space. Please refer to Table 1 for a comprehensive comparison of exteroception datasets.

### 2.2 QUADRUPED ROBOT PERCEPTION METHODS

As stated, this work focuses on quadruped exteroception tasks; proprioceptive methods for locomotion are omitted. Exteroceptive methods primarily evaluate environmental traversability for navigation and interaction, divided into terrain recognition and object recognition. Terrain recognition classifies ground surfaces to assess traversability, while object recognition detects obstacles (e.g., pedestrians, traffic cones) in 2D/3D space. Representative approaches include FLOBOT's SVM and Bayesian tracking (Yan et al., 2020), TBD Pedestrian's ByteTrack-based 2D tracking (Wang et al., 2024), L-CAS's LiDAR clustering with UKF tracking and SVM classification (Yan et al., 2018), JRDB's YoloV3/RetinaNet for 2D detection and Frustum PointNet for 3D detection (Martin et al., 2021), and SiT/CODa's LiDAR-based detectors (FCOS3D/PointPillar/CenterPoint) (Bae et al., 2023; Zhang et al., 2024).

Given L4Dog's focus on exteroception in complex urban roads with dense traffic, we adopt autonomous driving paradigms by formulating quadruped exteroception as Bird's-Eye-View (BEV) perception. Our technical approach combines whitelist-based BEV recognition and non-whitelist occupancy recognition methods.

### 2.3 BEV PERCEPTION IN AUTONOMOUS DRIVING

BEV perception has experienced significant advancements in the field of autonomous driving in recent years. The core concept involves mapping multi-sensor data through coordinate transformation to unify features in the BEV space for representation and learning. Representative works for BEV detection include LSS (Philion & Fidler, 2020), BEVFormer (Li et al., 2022), BEVDet (Huang et al., 2021), and BEVFusion (Liu et al., 2022). Occupancy Prediction, a novel benchmarking task

Table 1: Comparison of exteroception datasets for (quadruped) robots. L4Dog supports 360-degree panoramic sensor fusion recognition, outperforming SOTA benchmarks in data scale, 3D object annotation capacity, and object density. L4Dog enables multi-modal external perception tasks including BEV perception, trajectory prediction, and occupancy prediction.

| Dataset | Published | Num Samples/ Duration | Scene | 360° L+C/ Obj Density/ 3D objs | Exteroception Tasks | Sensors |
|---|---|---|---|---|---|---|
| NCLT Carlevaris et al. (2016) | IJRR IF 5.0 | N/A 34.9 h | indoor outdoor campus | ✓ N/A 0 | N/A | 32&2-beam LiDAR 360° Camera |
| L-CAS Yan et al. (2018) | IROS | 28,002 0.82 h | indoor office | ✗ 1.12 6140 | Human Detection & Tracking | 16-Beam LiDAR N/A |
| GoStanford Hirose et al. (2018) | IROS | 10,560 N/A | indoor office | ✗ N/A 0 | 2D Traversable Probability | N/A 360° RGB Camera |
| FLOBOT Yan et al. (2020) | ISR IF 4.3 | 16,570 0.46 h | indoor airport etc. | ✗ N/A 968 | Human Detection & Tracking | 16-Beam & 2D-LiDAR RGB-D Stereo |
| RECON Shah et al. (2021) | arXiv | 5,000 N/A | outdoor 9 sites | ✗ N/A 0 | N/A | 2D LiDAR Stereo Camera |
| JRDB Martin et al. (2021) | TPAMI IF 20.8 | 60,000 1.07 h | indoor outdoor campus | ✓ 30 1.8 million | Human Detection & Tracking | 2x 16-Beam LiDAR Stereo & Fisheye Camera |
| SCAND Karnan et al. (2022) | RA-L IF 5.3 | N/A 8.7 h | indoor outdoor campus | ✓ N/A 0 | N/A | 16-Beam LiDAR RGB-D & surround RGB |
| Seattle Shaban et al. (2022) | CoRL | N/A 1 h | outdoor offroad | ✗ N/A 0 | LiDAR Semantic segmentation | 64-Beam LiDAR |
| ViKiNG Shah et al. (2022) | RSS | N/A 12 h | outdoor sidewalks/parks | ✗ N/A 0 | N/A | N/A 170° RGB Camera |
| SACSoN Hirose et al. (2023) | RA-L IF 5.3 | N/A 75 h | indoor office | ✗ N/A 0 | N/A | 2D LiDAR Spherical RGBD |
| SiT Bae et al. (2023) | NeurIPS | 12,000 0.33 h | indoor outdoor Open Scenes | ✓ 26.7 0.32 million | Human Detection, Tracking, Prediction | 2x 16-Beam LiDAR 5x Camera |
| ForestTrav Ruetz et al. (2024) | IEEE Access IF 3.6 | N/A N/A | outdoor forest | ✓ N/A 0 | probabilistic 3D voxel map | 16-Beam LiDAR/ 4x RGB Camera |
| CEAR Zhu et al. (2024) | RA-L IF 5.3 | N/A N/A | indoor outdoor | ✗ N/A 0 | N/A | 16-Beam LiDAR/ Event&RGBD Camera |
| TBD Pedestrian Wang et al. (2024) | ICRA IF 4.55 | N/A 3.55 h | indoor Mall | ✓ N/A 11,716 | Human Tracking | 3D-LiDAR/ 360°&Stereo Camera |
| CODa Zhang et al. (2024) | T-RO IF 10.5 | 34,800 1 h | indoor outdoor campus | ✗ 37.4 1.3 million | LiDAR 3D Detection | 128-Beam LiDAR/ 2xRGB&RGBD Camera |
| QuadTrack Luo et al. (2025) | CVPR | 19,000 0.5 h | outdoor campus | ✗ 9.89 0.19 million | 2D mot | N/A/ Panoramic Camera |
| **L4Dog(ours)** | 2025 | **360,000** 10 h | outdoor **complex urban** | ✓ **48.7** **17.5 million** | **BEV Detction & Tracking Trajectory Prediction Occupancy Prediction** | 32-Beam LiDAR/ 5x RGB Camera |

introduced in autonomous driving, addresses the challenge of detecting non-standard obstacles by representing 3D space through a voxelized grid. Notable approaches include Occ3D (Tian et al., 2023), OpenOcc (Tong et al., 2023), SparseOcc (Liu et al., 2023), FBOcc (Li et al., 2023), and FlashOcc (Yu et al., 2023). This work builds on autonomous driving's BEV paradigm by introducing the first first-person BEV perception methodology for quadrupedal robots. The data distribution, scene complexity, and object representation challenges in this context differ significantly from autonomous driving, offering unique value for quadrupedal robotics. Beyond BEV detection and occupancy prediction, we propose BEV tracking and trajectory prediction tasks, introducing four novel external perception benchmarks tailored for quadrupedal robots. Finally, we introduce the first multi-task BEV recognition framework integrating all four benchmarks.

## 3 DATASET

### 3.1 PLATFORM & SENSORS

We selected a quadruped robotic dog as our data acquisition platform. Compared to alternative mobile robotic platforms (e.g., Clear Path robots), legged systems offer superior terrain adaptability, enhanced mobility, and greater commercialization potential. Specifically, to address diverse urban scenarios including sidewalks, roadways, and tactile paving (Section 3.3), we employed the wheeled

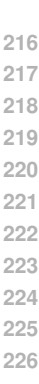
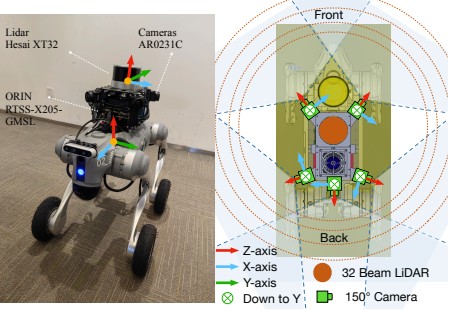

Figure 2: Sensor Setup on Unitree Go2. One 32-beam LiDAR and five 150°-FOV cameras. Blue: camera FoV; brown dots: LiDAR coverage.

Table 2: Sensor Specifications. We utilize 5 x cameras and 1 x LiDAR deployed in a 360°configuration. IMU uses the built-in IMU of the robotic dog.

| Sensor | Num | Specifications |
|---|---|---|
| Camera | 5 | RGB image @ 1920x1080 resolution, 10Hz, FOV=150°. |
| LiDAR-32 | 1 | Spinning, 32 beams, 10Hz, $360° \times 31$FOV @ $0.18° \times 1$ resolution, 0.05–120m range @ $\pm 0.5$cm accuracy, with up to 6.4M points per second. |

quadruped Unitree Go2 (Unitree, 2025-07-27) as the L4Dog acquisition platform. This hybrid locomotion system enables wheeled movement on flat surfaces and legged locomotion on uneven terrain (e.g., tactile paving) and elevation changes (curbs), demonstrating exceptional terrain traversal capabilities. For obstacle avoidance (vehicles, motorcycles, pedestrians, static obstacles) during urban navigation, precise 3D object recognition is essential.

Consequently, a high-performance 360° perception system was implemented, comprising one 32-beam LiDAR and five RGB cameras to provide fused point cloud and visual data. Compared to 16-beam LiDARs (Table 1), the 32-beam configuration yields higher point density and extended detection range. Unlike RGB-D or stereo cameras, the multi-camera panoramic system delivers comprehensive 360° visual coverage that spatially aligns with LiDAR point clouds, enriching 3D data with semantic information. Relative to panoramic cameras, this multi-camera configuration achieves superior detection range and reduced image distortion. The 360° LiDAR-camera fusion approach follows autonomous driving paradigms (nuScenes (Caesar et al., 2020), nuPlan (Caesar et al., 2021), Waymo (Sun et al., 2020), PandaSet (Xiao et al., 2021), Argoverse (Wilson et al., 2023)), addressing L4 perception challenges in complex pedestrian/vehicle environments while supporting BEV perception formulations. Wide-angle RGB cameras were mounted vertically (90° rotation) to maintain 360° coverage while expanding vertical perception. This configuration ensures full-body imaging of nearby pedestrians (0.4m). Sensor specifications are listed in Table 2.

### 3.2 COORDINATES, CALIBRATION AND SYNCHRONIZATION

#### 3.2.1 COORDINATE SYSTEMS

The L4Dog platform employs five coordinate systems for spatial perception and sensor fusion Figure 2, including image UV coordinates for 2D pixel representation in vision data, camera coordinates as a 3D frame centered at the optical axis for geometric transformations, IMU coordinates aligned with inertial sensor axes for motion state estimation, LiDAR coordinates for high-resolution 3D point cloud mapping, and robot coordinates as a body-fixed frame for navigation and control. These coordinate systems are synergistically integrated through transformation matrices, with cross-sensor calibration achieved via the following methods.

#### 3.2.2 CALIBRATION PROCEDURES

- **Cameras:** Calibrated using checkerboard patterns and pinhole camera models to establish image-to-camera coordinate transformations (Zhang, 1999).

- **Camera-to-LiDAR:** Extrinsic calibration performed pairwise, with projection matrices optimized until static point cloud projections achieved pixel-level alignment.

- **IMU-to-Robot:** Transformation derived from measured installation offsets and angles.

- **LiDAR Motion Compensation:** IMU motion estimates applied for dynamic point cloud distortion correction.

- **LiDAR-to-IMU:** LiDAR-to-IMU calibration is initialized using CAD drawings and on-site installation measurements, and further refined via the LI-Init calibration method (Zhu et al., 2022).

Table 3: Data collection scenes statistics.

| Scene ID | Scene | Clips | Annotated Objects | Features |
|----------|-------|-------|-------------------|----------|
| S001 | Subway | 363 | 1.93M | dense pedestrians |
| S002 | Traffic light intersections | 1798 | 10.3M | complex traffic flow |
| S003 | Open Road | 1081 | 3.5M | mix of pedestrians and vehicles |
| S004 | Tactile Paving | 358 | 1.75M | narrowly passable |

### 3.2.3 SYNCHRONIZATION

All sensors were synchronized via a high-precision Precision Time Protocol (PTP) server, with timestamps referenced to the LiDAR's timestamp. Camera exposure triggers were initiated at Li-DAR scan center alignment, defining camera timestamps. LiDAR timestamps marked completion of full rotational scans, with motion compensation applied using localization data to account for scan duration.

### 3.3 DATA COLLECTION SCENES AND COLLECTION PLANS

L4Dog focuses on first-person 3D perception datasets for quadruped robots operating in complex urban environments. The data was collected during peak hours in four distinct urban scenarios (Table 3): 1) Subway stations, characterized by high pedestrian density and dynamic interactions between people and non-motorized vehicles; 2)Traffic light intersections involving complex interactions between vehicles and pedestrians; 3) Open roads with mixed vehicle/non-motorized traffic; 4) Tactile paving areas with narrow space, requiring specialized navigation for assistive applications. Representative annotated samples are visualized in Figure 4.

### 3.4 GROUND TRUTH FORMATS AND ANNOTATION

As previously formulated, perception tasks are structured as BEV problems comprising: 1) BEV 3D detection ground truth for whitelisted objects; and 2) 3D traversability (occupancy) ground truth for non-whitelisted entities.

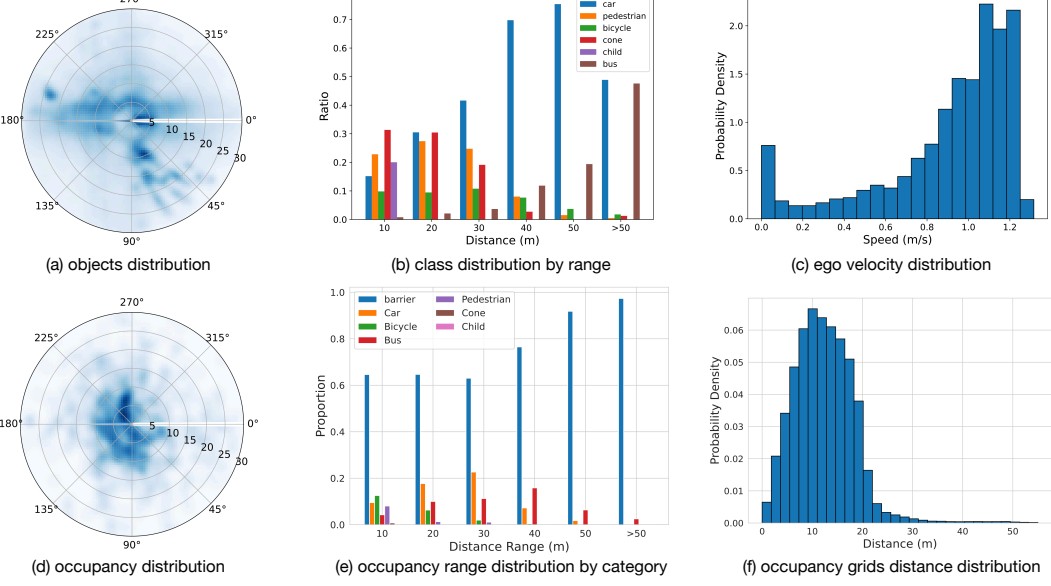

Figure 3: L4Dog Dataset Statistics: (a) 360° heatmap of object distribution relative to the ego robot. (b) Object category distribution by distance from the ego robot. (c) Ego robot speed distribution. (d) 360° heatmap of occupancy grid distribution relative to the ego robot. (e) Occupancy category distribution by distance from the ego robot. (f) Occupancy grid distance distribution. Note "child" and "leg" are treated as interchangeable terms for the same category.

### 3.4.1 BEV 3D OBJECT ANNOTATION

Each sensor frame was manually annotated for whitelisted objects within 50m. Annotations include 3D bounding boxes parameterized as (id, cls, x, y, z, w, l, h, yaw), where: *id* denotes unique object identifiers enabling tracking/prediction across 10Hz frames (100ms intervals); *cls* indicates object category $\in$ {car, bus/truck, pedestrian, cyclist, static obstacle, legs/child}; (x, y, z) specifies robot-centric coordinates (meters); (w, l, h) defines physical dimensions (meters); and yaw defines the heading angle. Notably, 'legs' category denotes pedestrians within 0.4m where upper-body occlusion prevents full-body detection. Annotations were performed using a custom LiDAR-RGB fusion tool, primarily labeling 3D boxes in point clouds with image projection validation. All annotations underwent secondary quality assurance, achieving $\geq$ 98% accuracy.

### 3.4.2 3D OCCUPANCY GRID ANNOTATION

Beyond dynamic object perception, L4Dog addresses non-whitelisted object recognition and 3D traversability estimation (e.g., curb negotiation, obstacle avoidance). Departing from elevation map representations ( (Miki et al., 2022)), we formulate this as 3D occupancy estimation, encoded as cls, x, y, z, grid_size to resolve traversability and open-set recognition.

Occupancy ground truth generation adapts autonomous driving methodologies (Tian et al., 2023; Tong et al., 2023): 1) Dynamic objects (Section 3.4.1 annotations) are transformed to object-centric coordinates for multi-frame point cloud accumulation; 2) After temporal reconstruction and motion compensation, static backgrounds are processed via multi-frame point cloud fusion and mesh reconstruction; 3) Ground planes (relative traversable surfaces) are segmented and removed; 4) Remaining point clouds are voxelized (grid_size=0.2m) within a cylindrical volume (radius=50m, height=[-1m,4m], robot-centric), into 8 categories (besides bbox categories, add free space and unknown). The ground truth labels underwent a final round of manual quality control and refinement. Occupancy representations are illustrated in Figure 1.

### 3.5 STATISTICAL ANALYSIS

Dataset statistics are presented in Figure 3, where we analyze the distribution of objects and occupancy grids under the ego robot. The analysis includes heat map distributions, category distributions within 10-meter intervals, distance distributions, and velocity distributions during robot data collection. As shown in the figure, L4Dog exhibits characteristics such as complex multi-class object distributions, long-range objects, and high-density occupancy grids.

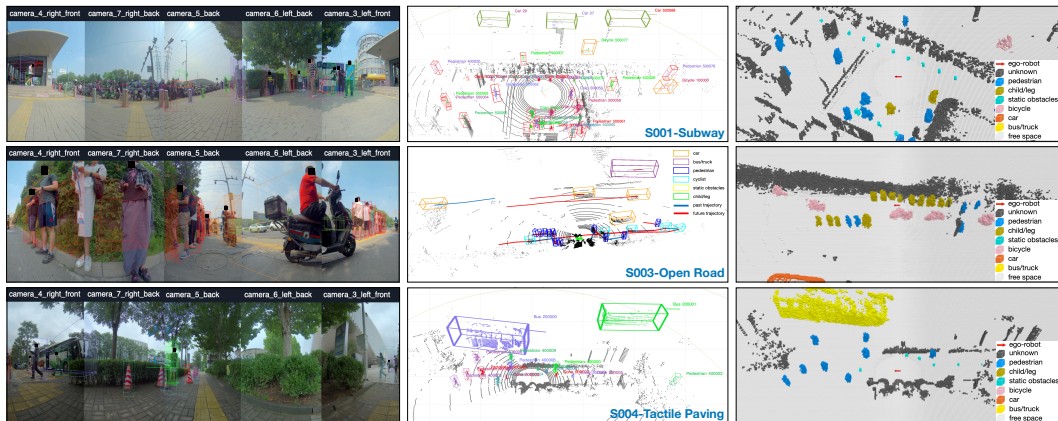

Figure 4: More collection scenes and GT illustration samples for each scene. For S002 traffic-light intersection please refer to Figure 1. From left to right: Cameras, LiDAR, Occupancy. We showcase detection & tracking frame in S001 and S004, motion frame in S003.

### 3.6 PRIVACY PROTECTION

All human faces and license plates in image data were anonymized using mosaic blurring to ensure privacy and data security.

## 4 BENCHMARKING & BASELINES

We introduce two core benchmarks on the L4Dog dataset: BEV Object Recognition (Section 4.1) and Occupancy Prediction (Section 4.2). The BEV Object Recognition benchmark encompasses three subtasks: BEV Object Detection (Section 4.1.1), BEV Object Tracking (Section 4.1.2), and Object Trajectory Prediction (Section 4.1.3). Furthermore, we propose a multitask baseline framework that enables simultaneous performance of all four perception tasks through a unified neural network architecture (Section 4.3). Notably, L4Dog represents the first and only work in quadruped robotics perception research to systematically establish benchmarks for both BEV object recognition and occupancy prediction.

### 4.1 BEV OBJECT RECOGNITION

#### 4.1.1 BEV OBJECT DETECTION

Task Description: Analogous to BEV detection in autonomous driving research, L4Dog's BEV object detection aims to identify object categories, positions, orientations, and dimensions within the robot's 360° surroundings using LiDAR and surround-view images. Distinct from automotive applications, L4Dog presents algorithmic challenges including dense non-rigid objects, severe pedestrian occlusion, and partial observation of pedestrians (leg categories) from the robot's low vantage point. The formulation is expressed as:(cls, conf, $x$, $y$, $z$, $w$, $l$, $h$, yaw) $= \mathcal{F}((I_0, I_1, I_2, I_3, I_4), L, t)$ where $I_i$ denotes JPEG images from five cameras, $L$ represents LiDAR point clouds (PCD format), and $t$ indicates the temporal component (optional for single-frame detection; required for 4D multi-frame detection). The outputs include object class ($cls$), confidence ($conf$), position $(x, y, z)$, bounding box dimensions $(w, l, h)$, and yaw angle. For evaluation, we employ widely used mAP as the metric: $\text{mAP} = \frac{1}{N} \sum_{c=1}^{N} \left( \frac{1}{11} \sum_{r \in \{0, 0.1, ..., 1\}} \max_{\tilde{r} \geq r} p_c(\tilde{r}) \right)$.

#### 4.1.2 BEV OBJECT TRACKING

Task Description: This task focuses on associating unique IDs to detected objects across consecutive frames. By integrating object detection, motion modeling, and data association techniques, it addresses challenges such as target occlusion and background interference. The benchmark supports both two-stage *tracking-by-detection* and one-stage *tracking-by-learning* paradigms. BEV object tracking further enables downstream functionalities including velocity/acceleration estimation and trajectory prediction. Performance is evaluated using MOTA:
$\text{MOTA} = 1 - \frac{\sum_t (\text{FP}_t + \text{FN}_t + \text{IDSW}_t)}{\sum_t \text{GT}_t}$.

#### 4.1.3
OBJECT TRAJECTORY PREDICTION

Task Description: This component predicts future motion trajectories of objects based on their historical movements (using Track IDs). Implementations may follow either two-stage *prediction-by-observation* or one-stage *learning-based*

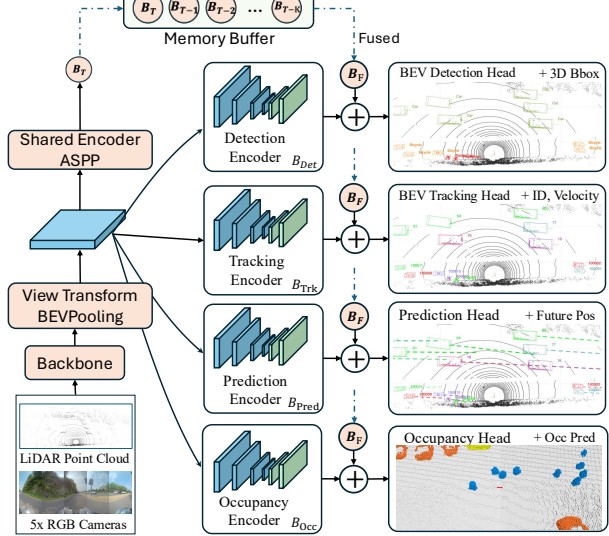

Figure 5: OmniBEV4D: Multitasking Baseline Method

approaches. Quantitative assessment uses ADE and FDE metrics: $\text{ADE} = \frac{1}{T \cdot N} \sum_{i=1}^{N} \sum_{t=1}^{T} \left\| \hat{\mathbf{p}}_t^{(i)} - \mathbf{p}_t^{(i)} \right\|$, $\text{FDE} = \frac{1}{N} \sum_{i=1}^{N} \left\| \hat{\mathbf{p}}_T^{(i)} - \mathbf{p}_T^{(i)} \right\|$.

### 4.2 OCCUPANCY PREDICTION

Task Description: This task predicts 3D spatial occupancy within the robot's sensing range to determine navigable areas. We discretize a cylindrical volume (radius: 50m; height: [-1m, 4m]) centered on the robot into grids (grid_size = 0.2m). Similar to autonomous driving formulations, each grid

is characterized by $\{cls, conf, x, y, z, occupied\}$, where *cls* denotes category, *conf* indicates confidence, $(x, y, z)$ represents robot-centric coordinates, and *occupied* is a binary occupancy flag. We utilize mIoU (Tian et al., 2023) and RayIoU (Liu et al., 2023) as metrics. mIoU measures voxel-wise overlap between predicted and ground-truth occupancies, while RayIoU evaluates occupancy consistency along sensor rays by comparing predicted and actual ray termination points.

### 4.3 METHOD & EXPERIMENTS

#### 4.3.1 BASELINE METHOD: OMNIBEV4D

A naive baseline approach would apply classical methods (e.g., BEVDet (Huang et al., 2021) for detection, ByteTrack (Zhang et al., 2022) for tracking) independently to each task. However, we contend that such single-task baselines offer limited value for L4Dog's complex scenarios, as no individual task suffices for quadruped robots' navigation requirements.

Moreover, combining four separate baselines incurs significant computational redundancy, precluding real-time deployment. Therefore, we propose OmniBEV4D—a strong multitasking baseline for L4Dog perception. As illustrated in Figure 5, this LiDAR-camera fusion network maximizes computational sharing through: 1) joint feature extraction from heterogeneous sensors, 2) unified feature fusion, and 3) shared 4D memory buffer. Task-specific heads then branch for BEV detection, tracking, trajectory prediction, and occupancy estimation.

#### 4.3.2 L4DOG EXPERIMENTS & ABLATIONS

We conduct quantitative evaluations of OmniBEV4D on the L4Dog dataset, presenting comparative results against classical quadruped exteroceptive methods across the proposed tasks (see Table 4). As demonstrated, OmniBEV4D achieves state-of-the-art performance while handling multiple tasks. Furthermore, we conducted ablation studies by training OmniBEV4D on nuScenes and CODa datasets, followed by evaluation on L4Dog. The performance degradation observed highlights the distinct distribution and complexity characteristics of our proposed L4Dog dataset.

Table 4: Quantitative Evaluation on L4Dog.

| Methods | BEVDet mAP↑ | BEVTrk MOTA↑ | TrajPred ADE/FDE↓ | OccPred mIoU/RayIoU↑ |
|---|---|---|---|---|
| PointPillar (Lang et al., 2019) | 58.4% | — | — | — |
| BEVFusion (Liu et al., 2022) | 70.1% | — | — | — |
| ByteTrack (Zhang et al., 2023) | — | 57.9% | — | — |
| GANet (Wang et al., 2022) | — | — | 1.24/2.16 | — |
| FBOcc (Li et al., 2023) | — | — | — | 48.5%/43.2% |
| OmniBEV4D-nuScenes | 52.4% | 58.3% | 1.45/1.98 | 33.2%/36.4% |
| OmniBEV4D-CODa | 60.3% | — | — | — |
| OmniBEV4D (ours) | 70.4% | 65.4% | 1.24/1.76 | 45.6%/52.4% |

## 5 CONCLUSION & FUTURE WORK

We present L4Dog, the largest and most complex exteroceptive perception dataset to date in quadruped robotics research. L4Dog encompasses temporally continuous multi-modal sensor data and human-annotated ground truth across complex urban scenarios including subway stations, traffic intersections, and open roads. This work pioneers the formulation of quadruped robotic perception as BEV tasks, establishing comprehensive benchmarks for BEV detection, BEV tracking, and trajectory prediction. Furthermore, we introduce the first occupancy prediction benchmark with corresponding ground truth for quadruped robots.

To address the multitasking requirements of practical autonomous navigation, we propose OmniBEV4D—a unified framework that simultaneously generates inference results for all perception tasks through shared spatiotemporal feature computation. Quantitative evaluations on the L4Dog dataset validate the effectiveness of the OmniBEV4D approach, with ablation studies highlighting the dataset's value.

Future work will incorporate natural language annotations to support Vision-Language Models (VLM) and Visual Question Answering (VQA) research, navigation trajectory recordings for Vision-and-Language Navigation (VLN) studies, and low-level control signal acquisition to enable Vision-Language-Action (VLA) research.

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
