# OpenReview forum: "L4Dog: Towards BEV Perception for Quadruped Robots in Complex Urban Scenes"
_ICLR.cc/2026/Conference — ICLR 2026 Conference Withdrawn Submission_

### Official Review · Reviewer_xKdU · 2025-10-23

**Soundness:** 3
**Presentation:** 1
**Contribution:** 2
**Rating:** 6
**Confidence:** 3

**Summary:**

This paper presents L4Dog, a large-scale exteroceptive 3D perception dataset designed for quadruped robots in complex urban environments. It offers 360° surround-view sensor data with manual annotations across diverse scenes and introduces a unified BEV-based multi-task perception framework (OmniBEV4D) for detection, tracking, prediction, and occupancy estimation. The work aims to bridge the gap between research and real-world deployment by providing a comprehensive benchmark for embodied intelligence in urban navigation.

**Strengths:**

•  The dataset is of impressive scale and diversity, covering highly complex urban scenarios.

•  Collecting and annotating such large-scale multimodal data is a major effort and can benefit future research in robotics.

•  The unified BEV-based formulation provides a coherent framework for multiple perception tasks, which may inspire follow-up work.

**Weaknesses:**

•  Although the paper emphasizes embodied intelligence, it primarily focuses on perception tasks. It would be valuable to include benchmarks or discussions related to Vision-and-Language Navigation (VLN) or Visual Question Answering (VQA) to demonstrate broader embodied reasoning capabilities.

•  The paper presentation could be improved: the use of \citet and \citep is wrong, and table captions are placed awkwardly, affecting readability.

**Questions:**

How does the proposed OmniBEV4D framework generalize across different robot platforms or sensor configurations?

---

> ### Author Response · Authors · 2025-11-17
> **VLN/VQA Integration, OmniBEV4D Performance, and Manuscript Improvements**
>
> We sincerely appreciate your recognition of our work. Below, we address your questions and suggestions.
>
> ### 1) Valuable if Include VLN or VQA
>
> Thank you for your valuable suggestion. Indeed, Vision-and-Language Navigation (VLN), Visual Question Answering (VQA), or Vision-and-Language Alignment (VLA) hold significant research value within the robotics domain. As mentioned in our Conclusion section, we have planned to extend our work in these areas as part of our future endeavors. (In fact, extensions to VQA and VLN based on the Chain-of-Thought framework have already been completed. Please stay tuned for our upcoming publications.)
>
> ### 2) Generalization Ability of OmniBEV4D Across Other Robotic Platforms or Sensors
>
> To further demonstrate the generalization capabilities of OmniBEV4D across different robotic platforms and sensor configurations, we have included additional perception experiments conducted on the JRDB and CODa datasets. The performance metrics are summarized in the table below:
>
> **Table 1. Performance Comparison on JRDB and CODa Datasets**
>
> | JRDB |                |                    | CODa |                |
> |---------------------|----------------|--------------------|----------------------|----------------|
> | Method              | mAP (IoU=.3)%    |                    | Method               | BEVAP (IoU=.7)% |
> | TANet[1,2]          | 53.87          |                    | CenterPoint[4]       | 82.08          |
> | DRFDFF[3]           | 76.28          |                    | PVRCNN[5]            | 92.08          |
> | OmniBEV4D           | 76.68          |                    | OmniBEV4D            | 93.45          |
>
> ### 3) Improve Presentation
>
> We apologize for any inconvenience caused by the readability issues in our manuscript. Your feedback is highly appreciated, and we will revise our manuscript according to your suggestions to improve its clarity and presentation.
>
> ## References
>
> [1] Liu, Zhe, et al. "Tanet: Robust 3d object detection from point clouds with triple attention." Proceedings of the AAAI conference on artificial intelligence. Vol. 34. No. 07. 2020.
>
> [2] Martin-Martin, Roberto, et al. "Jrdb: A dataset and benchmark of egocentric robot visual perception of humans in built environments." IEEE transactions on pattern analysis and machine intelligence 45.6 (2021): 6748-6765.
>
> [3] Guang, Jinzheng, et al. "DCCLA: Dense cross connections with linear attention for LiDAR-based 3D pedestrian detection." IEEE Transactions on Circuits and Systems for Video Technology (2024).
>
> [4] Yin, Tianwei, Xingyi Zhou, and Philipp Krahenbuhl. "Center-based 3d object detection and tracking." Proceedings of the IEEE/CVF conference on computer vision and pattern recognition. 2021.
>
> [5] Shi, Shaoshuai, et al. "Pv-rcnn: Point-voxel feature set abstraction for 3d object detection." Proceedings of the IEEE/CVF conference on computer vision and pattern recognition. 2020.

---

> ### Author Response · Authors · 2025-11-25
> **Improve Presentation**
>
> Dear Reviewer,
>
> We have revised the manuscript to correct the misuse of \citet and \citep and to improve the placement of table captions for enhanced readability.

---

> > ### Comment · Reviewer_xKdU · 2025-11-26
> >
> > Thank you for the clarification. My concerns have been resolved, and I will maintain my original score.

---

### Official Review · Reviewer_igbW · 2025-10-30

**Soundness:** 2
**Presentation:** 3
**Contribution:** 2
**Rating:** 4
**Confidence:** 3

**Summary:**

The paper introduces a dataset designed to evaluate BEV (Bird’s Eye View) perception for quadruped robots. Example BEV perception tasks include object detection, tracking, trajectory prediction, and occupancy prediction. The dataset contains both indoor and outdoor scenes and is substantially larger than those used in related studies.

The primary strength of this dataset lies in its scale. However, the dataset offers limitated coverage for complex terrain, which is a key capability of quadruped robots. Additionally, the paper does not clearly describe the identity anonymization procedures used in the dataset.

**Strengths:**

1. BEV perception for quadruped robots is an important research direction, and there is a clear need for a more comprehensive dataset in this area. The proposed dataset targests a key research gap in this domain.
2. In terms of scale, it is significantly larger than those presented in related works.

**Weaknesses:**

1. Missing complex terrain: If I understand correctly, the proposed dataset primarily contains planar terrain scenarios, both indoor and outdoor. However, it lacks coverage of challenging vertical terrains such as hills, slopes, stairs, and elevators. This omission is significant because one of the key advantages of quadruped robots over wheeled robots is their ability to navigate complex and uneven terrain. As a result, the dataset is limited in its ability to support comprehensive evaluation of future quadruped robot perception research.
2. Additionally, the paper does not clearly describe the identity anonymization procedures used in the dataset.

**Questions:**

1. It would be valuable to include a comparison of motion statistics with related datasets. For example, presenting pitch, roll, yaw, and acceleration curves from the quadruped robot’s motion sensors could help assess the complexity of the terrains captured in this dataset.
2. Furthermore, the data collection process may involve identity-related information such as children, human faces, or vehicle license plates. The paper should clarify how such sensitive information is handled during data collection and processing to ensure proper anonymization and ethical compliance.

**Details Of Ethics Concerns:**

The paper does not clearly describe the identity anonymization procedures used in the dataset.

---

> ### Author Response · Authors · 2025-11-17
> **Clarifications on Robot Platform, Data Anonymization, and Perception Scope**
>
> We sincerely appreciate the reviewer’s recognition of our work. Below, we provide clarifications in response to the comments.
>
> ### Clarification on Data Privacy and Robot Platform
>
> a) Section 3.6 of the manuscript explicitly describes our identity anonymization procedure. In accordance with privacy-preserving practices, all personally identifiable information—such as human faces and vehicle license plates—has been carefully anonymized via mosaic-style pixelation, as illustrated in Figures 1 and 4.
>
> b) As illustrated in Figure 2 of the paper, our robotic platform is a quadrupedal wheeled robot, which locomotes using four wheels mounted on a quadrupedal structure.
>
> ---
>
> ### 1) Missing Complex Terrain
>
> a) The core task of L4Dog is to enhance exteroceptive perception and autonomous navigation in complex urban road environments. Although the roads in these scenarios are primarily planar, they present extreme challenges in terms of dynamic obstacle density, occlusion, and heterogeneous traffic participants. Our key contribution is the formulation of the complex perception task for legged robots as a device-agnostic L4 perception task based on bird’s-eye-view (BEV) representations. Specifically, we address the following main perception challenges:
> (1) **Detection and tracking**: High-density pedestrians, cyclists, and large vehicles pose significant challenges to detection and tracking stability.
> (2) **Trajectory prediction**: Forecasting the future trajectories of all moving objects.
> (3) **3D occupancy understanding**: Interpreting the occupancy of cluttered scenes, including terrain, traversable areas, and static obstacles—elements that are both critical and common in urban environments.
>
> In these settings, the primary challenge is not slopes or stairs, but rather the complexity of the traffic environment. To the best of our knowledge, L4Dog is the first dataset to provide a large-scale benchmark for these key perception tasks in this context.
>
> b) The proposed robotic perception framework can also be extended to other complex terrains:
> i. **BEV Object Recognition**: Detecting and tracking other agents (e.g., pedestrians, vehicles) is essential regardless of terrain type—whether the robot operates on flat ground or on a slope.
> ii. **3D Occupancy Prediction**: This task predicts the 3D geometry and semantics of the surrounding space, which forms the basis for navigation on slopes, stairs, and other non-planar terrains. By providing the first high-quality occupancy ground truth for legged robots, our work establishes a prerequisite for addressing vertical terrain challenges. The perceptual output of our occupancy network can directly inform the motion planner about geometric features such as steps, impassable slopes, or stair structures.
>
> c) We acknowledge that complex terrain is indeed an important direction. In future data collection efforts, we plan to incorporate a wider variety of outdoor terrains and indoor scenes.
>
> ---
>
> ### 2) Identity Anonymization Procedure
>
> As stated in Section 3.6, human faces and license plates are anonymized using mosaic-style pixelation.
>
> ---
>
> ### 3) Include a Comparison of Motion Statistics with Related Datasets
>
> Our work focuses on BEV perception in urban scenes. Motion sensors are not involved in the BEV perception computation. However, we appreciate the reviewer’s valuable suggestion—motion sensors can indeed reflect the robot’s gait and pose during data collection—and we will consider this as future work.

---

> > ### Comment · Reviewer_igbW · 2025-11-25
> >
> > First, I sincerely apologize for missing the anonymization procedure in Sec 3.2. I will update my reviews and remove all comments related to anonymization.
> >
> > Second, regarding complex terrain and motion statistics, I agree that these are more suitable as future work. The proposed dataset in its current form already addresses the gap created by the lack of BEV data for quadruped robots in urban road environments. While I do believe that incorporating complex terrain, motion statistics, and VQA (as mentioned by Reviewer xKdU), would further strengthen the work, this does not diminish the contribution of the current paper. I would therefore like to raise my score to Accept.

---

### Official Review · Reviewer_adqi · 2025-11-03

**Soundness:** 3
**Presentation:** 3
**Contribution:** 3
**Rating:** 8
**Confidence:** 3

**Summary:**

This paper presents a large-scale BEV perception dataset for quadruped robots in complex urban scenarios, featuring high-quality manual annotations. A benchmark and baseline framework tailored for the evaluation and modeling of simultaneous multiple BEV perception tasks is concomitantly established. The paper is also the first one to contribute 360-degree occupancy annotations with a corresponding network.

**Strengths:**

1. To address the limitation of perception data for quadruped robotics in complex real-world urban scenarios, a large-scale 3D perception dataset with high-quality manual annotations for multiple BEV tasks is established.
2. The proposed dataset contains high-quality 360-degree 3D occupancy annotations for fine-grained quadruped robotics perception, with a corresponding occupancy prediction network.
3. A multi-task framework with shared temporal BEV space for multiple BEV perception tasks is developed.
4. Experimental results demonstrate the effectiveness of the multi-task framework.

**Weaknesses:**

1. Concern about the generalization ability. As shown in Fig. 3(b), the number of classes is fixed to several, which may hinder the generalization to objects beyond the predefined categories.
2. What's the detailed architecture of the proposed OmniBEV4D?

**Questions:**

1. What are the actual computational costs of OmniBEV4D, in terms of memory consumption, latency, etc?

---

> ### Author Response · Authors · 2025-11-17
> **Clarifications on Category Generalization, OmniBEV4D Architecture, and Performance Metrics**
>
> We sincerely appreciate your recognition of our work and thank you for your insightful feedback. In the following, we provide detailed responses to your concerns.
>
> ---
>
> ### 1) Concern Regarding Category Generalization Capability
>
> Fixed-category BEV detection inherently limits a model’s ability to generalize to previously unseen object classes. This limitation, however, serves as a primary motivation for incorporating the **3D occupancy prediction** task into our framework. Specifically, occupancy prediction introduces a generic **“barrier” class** that captures the presence of arbitrary obstacles—regardless of semantic category—in open-world environments. This design effectively circumvents the rigid, closed-set constraints imposed by conventional detection pipelines that rely on predefined category whitelists. Consequently, the **synergistic integration** of category-aware BEV detection and category-agnostic occupancy prediction enables robust perception in complex, dynamic, and open-world urban scenarios.
>
> ---
>
> ### 2) Detailed Architecture of OmniBEV4D
>
> **OmniBEV4D** is a multi-modal, multi-task **4D (3D + time) bird’s-eye-view (BEV) perception network** that fuses LiDAR and camera inputs. Its architecture is summarized as follows:
>
> - **Backbone**: A shared ResNet extracts features independently from five surround-view RGB images and voxelized LiDAR point clouds.
> - **Lift-and-Fuse**: Image features are lifted into 3D space using the **Lift-Splat-Shoot (LSS)** paradigm and then deeply fused with LiDAR voxel features within the BEV representation space.
> - **Contextual Encoding**: The fused BEV feature volume is processed by an **ASPP (Atrous Spatial Pyramid Pooling)** module to capture multi-scale spatial context.
> - **Temporal Aggregation**: Multi-frame BEV features are integrated via channel-wise fusion to construct a spatiotemporal representation.
> - **Task-Specific Refinement**: This spatiotemporal BEV tensor is further enhanced by task-specific encoder branches—specifically, **Region Proposal Network (RPN)-style encoders**—to generate a unified, task-aware feature map.
> - **Output Heads**: The final representation is routed to dedicated task heads for **joint prediction** of BEV object detection, tracking, and 3D occupancy.
>
> This design enables end-to-end learning of rich, temporally consistent BEV representations suitable for real-time robotic navigation.
>
> ---
>
> ### 3) Computational Cost of OmniBEV4D
>
> The inference latency and memory consumption of OmniBEV4D are benchmarked on two representative hardware platforms:
>
> | Model       | NVIDIA GeForce RTX 3080 Ti (FP16) | NVIDIA AGX Orin (FP16) |
> |-------------|-----------------------------------|---------------------|
> | OmniBEV4D   | 16 ms / 2750 MB GPU memory        | 75 ms / 2900MB GPU memory |
>
> These results demonstrate that OmniBEV4D achieves a favorable balance between perceptual richness and real-time feasibility on both desktop and embedded automotive-grade hardware.

---

> > ### Comment · Reviewer_adqi · 2025-11-21
> > **Official Comment**
> >
> > Thanks for the authors' responses. The issues have been addressed. This work offers an invaluable dataset, benchmark, and baseline framework to the field of quadruped robotic perception in complex urban scenes, inspiring researchers in related topics to develop advanced methods.

---

### Note · Authors · 2026-01-30

**Comment:**

We strongly condemn the Area Chair’s decision to reject a submission that received reviewer scores of 8, 8, and 6—scores and accompanying reviews that were finalized and updated on November 25. We find the Area Chair’s conduct deeply unprofessional and irresponsible: the meta-review appears to be a mere copy-paste of the reviewers’ initial assessments, entirely disregarding the substantive rebuttal we provided, as well as the clear consensus among all reviewers regarding the scientific merit and contribution of our work. Furthermore, we express our profound disappointment at the apparent inaction of the ICLR organizing committee and Program Chairs in response to our formal concerns regarding this decision. Below is the letter to PCs that never got replied.

Dear Program Chairs,

We are the authors of Paper #6807 (https://openreview.net/forum?id=BxGrZFfTp5). This letter constitutes a formal complaint against Area Chair qxkY for the unprofessional, negligent, and demonstrably unfair handling of our submission.

Our paper initially received a score of 846. During the rebuttal period, we actively and thoroughly addressed all concerns raised by the reviewers. By November 25—two days before the OpenReview system vulnerability incident on November 27—the paper’s score had been updated to 886 and remained stable thereafter. Critically, Reviewer igbW25 explicitly revised his/her assessment in writing, stating the decision should be upgraded to “Accept” (8), and this comment remained visible throughout.

Despite this clear consensus among reviewers post-rebuttal, Area Chair qxkY disregarded the substantive content of our rebuttal and the updated reviewer positions entirely. Instead, he/she based his/her final decision solely on the original weaknesses—weaknesses that had been fully addressed and acknowledged by the reviewers during rebuttal. Moreover, it appears that AC qxkY mistakenly believed that the score of 4 from Reviewer igbW25 was the updated rating, rather than the initial one, further demonstrating a fundamental failure in due diligence.

We strongly protest this unjust treatment and demand an immediate, fair re-review of our paper based on the actual, final opinions of the reviewers—not on the AC’s misinterpretation or oversight.

While we acknowledge that ICLR, like many others, was adversely affected by the OpenReview vulnerability, authors must not bear the consequences of an Area Chair’s careless and unprofessional conduct. We expect the Program Chairs to uphold the integrity of the review process and ensure that decisions reflect the true consensus of the reviewing committee.

Sincerely,
Authors of Paper #6807

**Withdrawal Confirmation:**

I have read and agree with the venue's withdrawal policy on behalf of myself and my co-authors.

---

### Meta-Review · Area_Chair_qxkY · 2026-01-06

**Summary:**

The paper introduces "L4Dog," a large-scale 3D perception dataset for quadruped robots in urban environments, along with a baseline multi-task model, OmniBEV4D. While reviewers appreciated the scale of the dataset and the effort in data collection, the paper is rejected due to a critical misalignment between the proposed platform (quadruped robots) and the environment (mostly planar urban roads). The unique value of legged robots lies in traversing complex, unstructured terrains (e.g., stairs, rubble), yet the dataset focuses on flat scenarios that do not distinguish the perceptual challenges from standard autonomous driving datasets. Furthermore, the proposed baseline relies on established architectures without significant methodological novelty, and the benchmarks lack higher-level "embodied intelligence" tasks (e.g., VQA/VLN) claimed in the motivation. The contribution is thus viewed as a valuable engineering resource more suitable for a specialized robotics venue than a representation learning conference. Thus, this paper was not recommended for acceptance.

**Reviewer Concerns:**

Addressed Concerns: (1) Implementation Details: The authors provided the requested architectural details of the OmniBEV4D baseline and computational cost metrics (latency/memory) to satisfy Reviewer adqi. (2) Privacy and Anonymization: The authors clarified the pixelation procedures for faces and license plates. Reviewer igbW acknowledged missing this in the initial read and considered this resolved. (3) Presentation: Issues regarding citation formatting and table caption placement raised by Reviewer xKdU were corrected in the revision.

Outstanding Concerns: (1) Platform-Environment Mismatch (Critical): Reviewer igbW initially flagged the "Missing complex terrain" as a significant weakness. While the reviewer accepted the authors' promise to include this in future work, the Area Chair maintains this as a critical, outstanding flaw. A dataset specifically pitched for quadruped robots that contains primarily planar, urban roads (accessible to standard wheeled robots) fails to justify its primary motivation. (2) Scope of "Embodied Intelligence": Reviewer xKdU noted the lack of VQA or VLN tasks, which contradicts the paper's "Embodied Intelligence" keywords and framing. The authors acknowledged this is future work, leaving the current submission limited to standard perception tasks, falling short of the broader claims in the abstract. (3) Generalization: Reviewer adqi's concern about the fixed category list limiting generalization remains valid, despite the authors' argument that the occupancy task mitigates this.

**Reviewer Scores:**

Reviewer adqi: 8 (Unchanged)

Reviewer igbW: 4 (Increased)

Reviewer xKdU: 6 (Unchanged)

---

### Decision · Program_Chairs · 2026-01-26

Reject